# Reproducibility Report
# SAdam: A Variant of Adam for Strongly Convex Functions

## Reproducibility Summary

**Scope of Reproducibility**

The central claim of the original paper is that a modification of Adam to utilize strong convexity yields faster convergence owing to a data dependent logarithmic regret bound, much akin to the recently proposed SC-RMSProp. Additionally the authors also claim that the higher performance is sustained even in non-convex optimization problems. We run the same experiments as the ones listed in the original paper, trying to reproduce the observations using a different framework (PyTorch) as compared to the authors (TensorFlow).

**Methodology**

We did not use any of the source code provided, preferring to implement everything from the scratch, however we did run the source code to verify the results, using the code provided itself. Additionally we performed a grid search over all the hyperparameters to get the optimal values of the same for each of the datasets. We used the NVIDIA Tesla K80 GPU provided on the Google Colab platform to perform all our experiments.

**Results**

All the claims of the paper, are in line with our empirical observations as well, and our results are within a range of 1-2 % of accuracy. For one of the experiment listed in the appendix of the original paper involving training of ResNet-18 with all of the optimizers, our results deviate. However, the same does not discredit any of the claims made by the original paper, as that additional experiment merely was shown to prove an additional benefit of using SAdam in the original paper.

**What was easy**

All experiments were easy to setup, from procuring all the 3 datasets to the selection of the hyperparameters. In addition, none of the experiments had a high runtime requirement, hence no additional hardware was required to conduct the experiments and training on Google Colab was sufficient.

**What was difficult**

Implementing the optimizers against which SAdam was to be compared took up a lot of time and compounded the work at hand. Besides that a few contradictions in the proposed values of the mini-batch size by the authors of SC-RMSProp and SAdam caused a delay in the completion of the reproducibility study.

**Communication with original authors**

The authors shared their codebase with us in our very first correspondence. We reached out to them on multiple occassions for clarifications, and they were always prompt in answering our doubts.

# 1   Introduction

Any variation with respect to Adam with a solid mathematical backing deserves a serious look and hence SAdam [1] got our attention. SAdam is an online convex optimizer that enhances the Adam algorithm by utilizing strong convexity of functions wherever possible. Although the motivation behind making these modifications are to improve performance in only convex cases, they prove to be effective even in non-convex cases. The Online Convex Optimization (OCO) framework involves iterative improvement of the model over a sequence of rounds. The OCO framework models the decision set as a convex set in Euclidean space denoted $K \subset R^n$. In each round $t$ the player (our model) chooses a decision $x_t$ from $K$. At the same time, the adversary reveals a cost function $f_t(.)$ and the player incurs a cost of $f_t(x_t)$. The difference between the cost incurred by the player and the cost incurred for taking the best decision is called Regret, and the minimization of this value entails improvement of the model.

$$R(T) := \sum_{t=1}^{T} f_t(x_t) - \min_{x \in K} \sum_{t=1}^{T} f_t(x) \tag{1}$$

Intuitively, an algorithm performs well if its regret is sublinear as a function of $T$ where $T$ is the time horizon i.e. $R_{Algo}(T) = o(T)$, and as the complexity of the regret decision decreases, the performance of the optimization increases.

Adam dynamically adjusts the step size and the update direction by exponential average of the past gradients. Although Adam has been extremely successful in many applications, it suffers from the non-convergence issue. AMSgrad and AdamNC were introduced to address these issues. These variants boast a data-dependant regret bounds that are $O(\sqrt{T})$ in the worst case.

SAdam is the first variant of Adam adapted to strongly convex functions. It follows the general framework of Adam, yet keeping a faster decaying rate step size controlled by time-variant hyperparameters to exploit strong convexity. Although it must be mentioned that similar ideas have successfully been applied in the past to the frameworks of Adagrad and RMSProp to get the variants called SC-Adagrad and SC-RMSProp [2]. Theoretical analysis of SAdam show a data-dependent $O(logT)$ regret bound for strongly convex functions, which means that it converges faster than AMSgrad and AdamNC in such cases.

# 2   Scope of reproducibility

We extrapolate the following claims from the original paper -

- Modification of Adam (henceforth referred to as SAdam) to utilize strong convexity yields faster convergence owing to a data dependent logarithmic regret bound, much akin to the recently proposed SC-RMSProp
- SAdam ends up outperforming all the existing optimization techniques in the context of deeper network training as well, which is inherently a non-convex optimization problem.
- A specific configuration of hyperparameters in SAdam generates a data-dependant logarithmic regret bound for SC-RMSprop algorithm.

We seek to verify all the claims listed above empirically replicating the experiments of the authors. We skip over the theoritical nuances of all these claims since the base paper provides detailed analysis replete with self-sufficient proofs.

# 3   Methodology

The authors made their source code available to us, upon request (now available on the OpenReview Portal as well). Their experimental setup made use of the Tensorflow framework, whilst utilizing the publicly available implementations [3] of the other optimizers against which comparisons were made. However we implemented the codebase from scratch using the PyTorch 1.7 framework, which meant reimplementing the other optimizers in question along with SAdam, due to a lack of a publicly available implementation of the same in PyTorch. All experiments were carried out on Google Colab with the NVIDIA Tesla K80. The following experiments were conducted -

- Calculation of Regret for L2 Regularized Logistic Regression on MNIST, CIFAR10, CIFAR100 for our pool of optimizers, to check the validity of claim 1.
- Computing test accuracy and training loss for a 4-layer CNN and ResNet18 on MNIST, CIFAR10, CIFAR100. for our pool of optimizers to check the validity of claim 2.
- Training a Multi-Layer LSTM on the PennTreeBank dataset on the Language Modelling task, to test the performance of SAdam in a context different from the usual vision domains it has hitherto been tested on.

## 3.1 Model Description & Experimental Setup

All of the optimizer algorithms try to learn the parameter vectors $\{w_i, b_i\}_{i=1}^{K}$, by minimizing the loss accumulated over each mini-batch of training samples $\{(x_i, y_i)\}_{i=1}^{m}$, where $y_i \in [K], \forall i \in [m]$, in each round $t$. This loss function also called the Cross Entropy Loss [4] has been used across all our experiments.

$$J(w,b) = -\frac{1}{m}\sum_{i=1}^{m} log\left(\frac{e^{w_{y_i}^T + b_{y_i}}}{\sum_{j=1}^{K} e^{w_j^T + b_j}}\right) + \lambda_1 \sum_{k=1}^{K} \|w_k\|^2 + \lambda_2 \sum_{k=1}^{K} b_k^2 \tag{2}$$

### 3.1.1 L2 Regularised Softmax Regression

A sequential layer is added and the outputs of the same are directly fed into the loss function. It must be noted that the PyTorch implementation of Categorical Cross Entropy Loss inherently applies Softmax, while computing the loss, hence to leverage their implementation we remove the last softmax function from our model, and directly feed in the fully connected layers output to the Loss function. We report the regret for this machine learning model across all our datasets and optimizers, considering the expert model of choice, to be the one trained by the Adam optimizer. This choice is completely trivial, and a different choice of an expert model would lead to a quantitative change of results, but would have no effect on the inferences obtained from the experiments.

### 3.1.2 4 Layer CNN

To verify the second claim, pertaining to the efficacy of SAdam in non-convex cases, we test the performance of the optimizers on a custom $4$ layer CNN, consisting of two convolutional layers (each with $32$ filters of size $3X3$), one max-pooling layer (with a $2X2$ window and $0.25$ dropout), and one fully connected layer (with $128$ hidden units and $0.5$ dropout). We employ $ReLU$ function as the activation function for convolutional layers and softmax function as the activation function for the fully connected ones. We train this network on all of our datasets, using all of the optimizers.

### 3.1.3 ResNet 18

To further substantiate the claim of the authors, we choose a deeper usage of architecture to see, whether the higher levels of performance are sustained as the number of layers increase. The popular ResNet-18 architecture was used for this purpose, the details of which can be found in the base paper [5]. To verify the claims made by the authors, we conduct the same experiment as them, running our models 10 times and finally taking a mean of the aggregate results on the CIFAR 10 dataset.

### 3.1.4 Multi-Layer LSTM

To test the performance of SAdam in unchartered territories, since it has hitherto only been used in vision problems, we used SAdam to train a language model on the PennTree bank dataset. Both a regularised and non-regularised RNN model containing 2 layers of LSTM was trained. For the regularised version, 650 hidden units were used in each layer of the LSTM, with dropout set at 0.5 and the model was trained for 39 epochs, whilst for the non-regularised model, the number of hidden units chosen was 200 and the model was trained for 13 epochs. In both cases, the LSTMs are unrolled over 35 steps, and trained with a batch size of 20, while for both vaildation and testing the batch size is set as 10. A learning rate scheduler is also used to aid in training, which decays the learning rate by a factor of 1.2 after every 6 steps. The model configuration and the experimental setup has been used as a basis for assessing the performance of a new optimizer as can be seen in the paper for AdaBelief.[6]

## 3.2 Datasets

The following datasets were used across experiments -

- MNIST : Dataset containing 60000 28X28 training images and 10000 test images depicting handwritten digits spread evenly across each digit (0-9). [7]
- CIFAR10 : A subset of the 80 million tiny images dataset and consists of 60,000 32x32 color images containing one of 10 object classes, with 6000 images per class. It was collected by Alex Krizhevsky, Vinod Nair, and Geoffrey Hinton.[8]
- CIFAR100 : Subset of the same dataset as CIFAR10, containing 600 images (500 for training and 100 for testing) per class for 100 classes. The 100 classes in the CIFAR-100 are grouped into 20 superclasses.[8]

- PennTreeBank : The Penn Tree Bank dataset maintained by the University of Pennsylvania contains text spanning around 2400 Wall Street Journal material along with a fully tagged Brown Corpus, and has over 4 million annotated words over a 10000 worded vocabulary. For the language model task, a 929K worded training set, 73k worded validation set, and 82k worded test is provided. [9]

## 3.3 Hyperparameters

An exhaustive list of hyperparameter settings used across optimizers and experiments have been surmised in table 1. In the table $\alpha$ refers to the learning rate to be used, $t$ refers to the step number, and thus in all convex cases, the learning rate is replaced by a degrading learning rate to account for the convergence. The $\beta_1$ and $\beta_2$ parameters appears in Adam and all of its variants controls the gradient moment decay rates as mentioned in [10]. The $\lambda_1$ and $\lambda_2$ parameters control the regularization in the cross entropy loss function, and is used only in convex cases. $\delta$ refers to a scalar value, that helps in avoiding divide by zero errors in optimizer updates. However, in the SC variants of RMSProp and Adagrad a time variant $\delta_{t,i}$ is used with the hyperparameters $\xi_1$ and $\xi_2$ controlling the variance. For SAdam, the optimal values for some of the hyperparameters mentioned in table 1 differ from experiment to experiment, and have been explored deeply in the subsequent section.

| Optimizer | Type | $\alpha$ | $\delta_{t,i}$ | $\beta_1$ | $\beta_2$ | $\lambda_1$ | $\lambda_2$ | $\xi_1$ | $\xi_2$ |
|---|---|---|---|---|---|---|---|---|---|
| OGD | Convex | 0.01 | $10^{-8}$ | — | — | 0.01 | 0.01 | - | - |
| | Non Convex | $0.01/t$ | $10^{-8}$ | - | - | 0 | 0 | - | - |
| Adam | Convex | 0.001 | $10^{-8}$ | 0.9 | 0.999 | 0.01 | 0.01 | - | - |
| | Non Convex | $0.001/t$ | $10^{-8}$ | 0.9 | 0.999 | 0 | 0 | - | - |
| SAdam | Convex | 0.01 | $10^{-2}$ | 0.9 | $1 - 0.9/t$ | 0.01 | 0.01 | - | - |
| | Non Convex | $0.01/t$ | $10^{-2}$ | 0.9 | $1 - 0.9/t$ | 0 | 0 | - | - |
| AmsGrad | Convex | 0.001 | $10^{-8}$ | 0.9 | 0.999 | 0.01 | 0.01 | - | - |
| | Non Convex | 0.001 | $10^{-8}$ | 0.9 | 0.999 | 0 | 0 | - | - |
| SC RMSProp | Convex | 0.01 | $\xi_1 e^{\xi_2 t V_{t,i}}$ | $1 - 0.9/t$ | - | 0.01 | 0.01 | 0.1 | 1 |
| | Non Convex | $0.01/t$ | $\xi_1 e^{\xi_2 t V_{t,i}}$ | $1 - 0.9/t$ | - | 0 | 0 | 0.1 | 0.1 |
| SC Adagrad | Convex | 0.01 | $\xi_1 e^{\xi_2 t V_{t,i}}$ | - | - | 0.01 | 0.01 | 0.1 | 1 |
| | Non Convex | $0.01/t$ | $\xi_1 e^{\xi_2 t V_{t,i}}$ | - | - | 0 | 0 | 0.1 | 0.1 |

Table 1: Value of hyperparameters used across experiments and optimizers

### 3.3.1 Hyperparameter Search

We use a grid search to determine parameters that provides the best performance for the algorithms in the regret computation experiment. We choose $\beta_1$ from $\{0.0, 0.9\}$. Taking $\beta_2 = 1 - \frac{\gamma}{t}$ we choose $\gamma$ from $\{0.9, 0.99, 0.999, 0.9999\}$. The learning rate $\alpha$ is chosen from $\{0.1, 0.01, 0.001, 0.0001\}$. These are taken as these were the values used by Kingma et al. [10] while testing the original Adam optimizer and used by some subsequent derivative papers as well [11]. The Figures below show the regret values for each parameter configuration averaged over 5 random seeds, individually for our vision datasets.

| Dataset | Learning Rate | $\beta_1$ | $\gamma$ |
|---|---|---|---|
| MNIST | 1e-2 | 0.9 | 0.9999 |
| CIFAR10 | 1e-3 | 0 | 0.9999 |
| CIFAR100 | 1e-3 | 0 | 0.999 |

Table 2: Optimal Hyperparameters for each Dataset

As reported in the table above, the best hyperparameter configuration per experiment differs from the corresponding values reported in the base paper. It must be noted though that the results for the optimal parameters are not overwhelmingly better than the values recommended. Fine tuning the hyperparameters in some of the experiments might lead to slightly better results, but the recommended values of 0.9 and 0.9 for $\beta_1$ and $\gamma$ respectively, irrespective of the experimental setup would lead to results closer to the best values possibly attainable using SAdam as an optimizer.

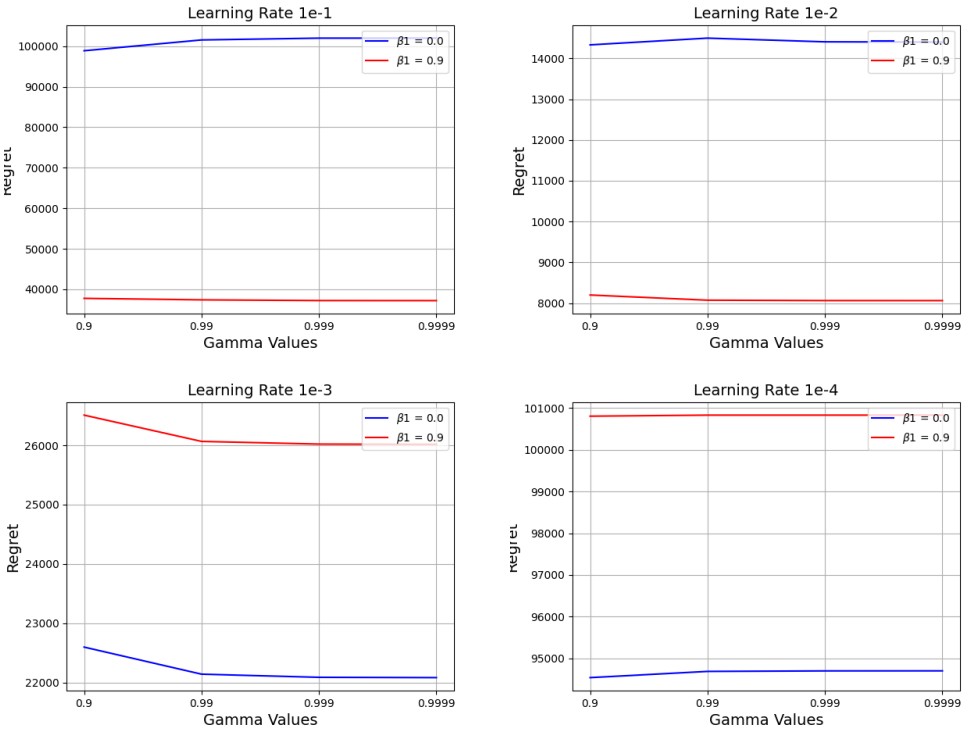

Figure 1: Hyperparameter Grid Search for MNIST on Regret calculation using L2-regularized Regression

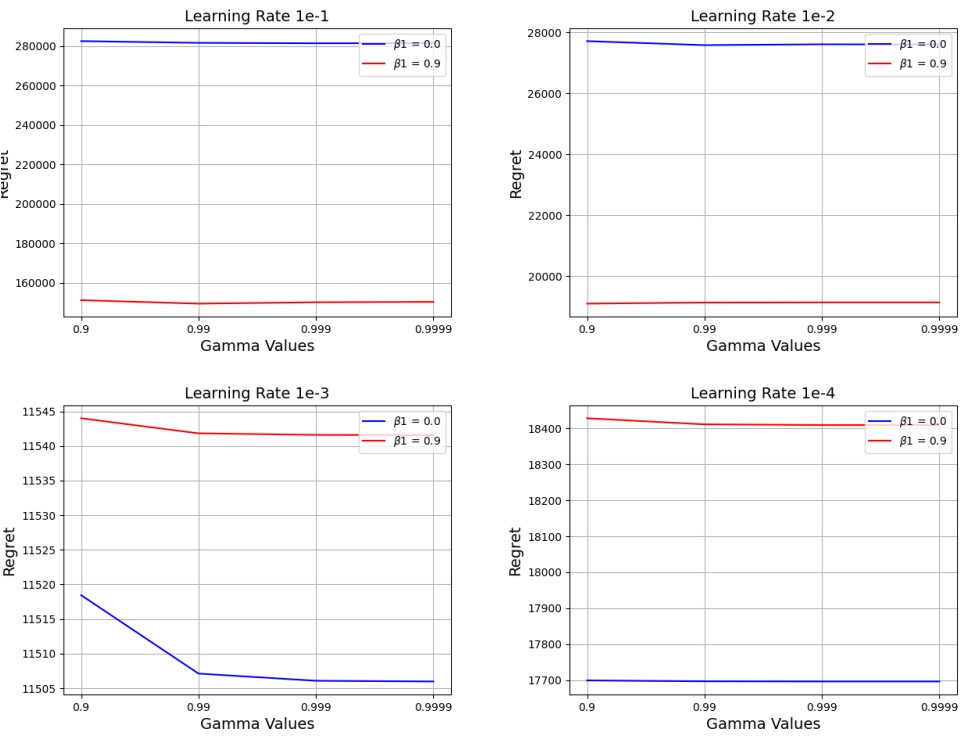

Figure 2: Hyperparameter Grid Search for CIFAR10 on Regret calculation using L2-regularized Regression

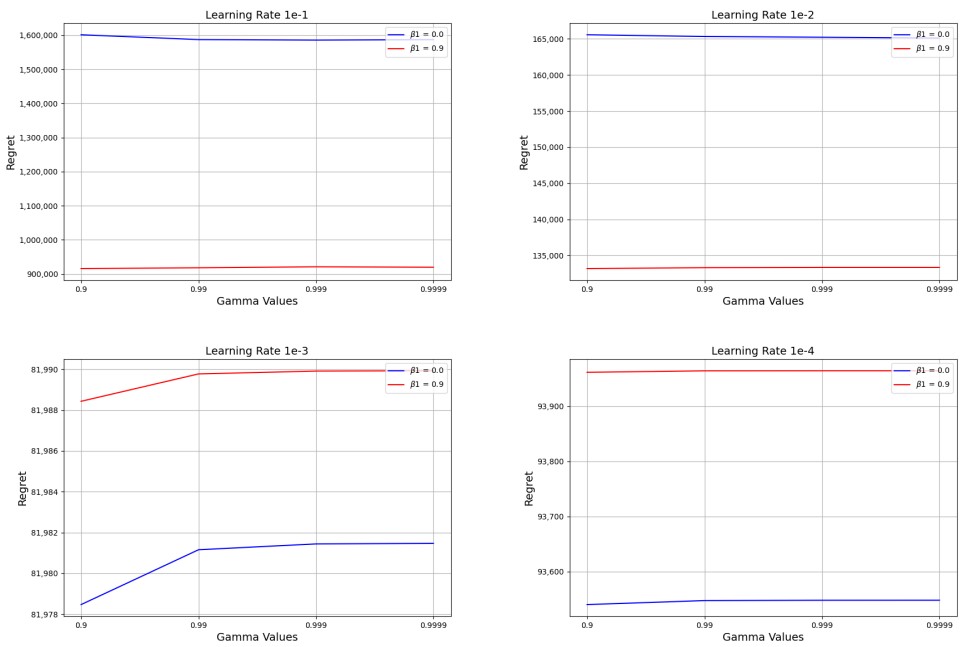

Figure 3: Hyperparameter Grid Search for CIFAR100 on Regret calculation using L2-regularized Regression

## 3.4 Computational requirements

All the experiments listed in the paper were carried out on the Google Colab platform, with the hardware acceleration provided by NVIDIA Tesla K80 GPU. As none of the experiments took more than 3 hours to completion, no cost had to be incurred on our part for conducting any of the experiments. The list of average runtimes across experiments NVIDIA Tesla K80 GPU can be seen in the table below -

| Model | Dataset | Batch Size | Epochs | Average Runtime (mins) |
|---|---|---|---|---|
| Softmax Regression | All | 60 | - | 15 |
| 4 layer CNN | MNIST | 64 | 50 | 13 |
| 4 layer CNN | CIFAR10 | 64 | 100 | 27 |
| 4 layer CNN | CIFAR100 | 512 | 100 | 52 |
| ResNet 18 | CIFAR10 | 64 | 100 | 128 |
| Non-Regularised 2 Layer LSTM | PennTreeBank | 20 | 39 | 110 |
| Regularised 2 Layer LSTM | PennTreeBank | 20 | 13 | 40 |

Table 3: Comparison of run times across experiments

The calculation of regret in the case of softmax regression warrants the availability of an expert model. Thus, the majority of the runtime is spent in obtaining this model, which once ready enables quicker completion of the regret calculations, since in that case, all of the other optimizers will only have to be trained over 1 epoch. After the trained model is avaiable, calculation of regret on a CPU takes an average of 3 - 4 minutes per experiment, and if the same is done over a GPU, the experiment would take around 40 - 50 seconds on an average.

## 4 Results

All the claims of the paper, are in line with our empirical observations as well. The original paper also compared the performance of SAdam to that of AdamNC, however we skip that particular comparison, as AMSGrad already serves as an extension for the Adam algorithm to improve upon the convergence of its optimization process, and the inclusion of AdamNC does not enrich the comparisons with any new information, with its performance almost being a replica of

AMSGrad and Adam. Aside from that fact, we were able to reproduce all of the experiments listed in the paper within a range of 1-2 % of accuracy in PyTorch.

## 4.1 Regret Calculation for L2-regularized Logistic Regression

This particular experiment seeks to establish the first claim made by the authors, pertaining to the utilisation of the convexity of the problem statement, to yield faster convergence. Across all the datasets in question, SAdam does converge fastest, and hence has the lowest overall regret. SC-RMSProp and SC-Adagrad have almost identical performances with very minor quantitative differences only in the decimal places. It must be noted that the final values of regret here are higher than the values reported in the actual paper due to the values in the paper being normalized by the batch size. Adam and AMSgrad also have identical performances, however their overall regret is much higher compared to the algorithms that take advantage of the convexity. OGD due to its simplistic algorithmic setup has the highest overall regret as expected in all of the datasets.

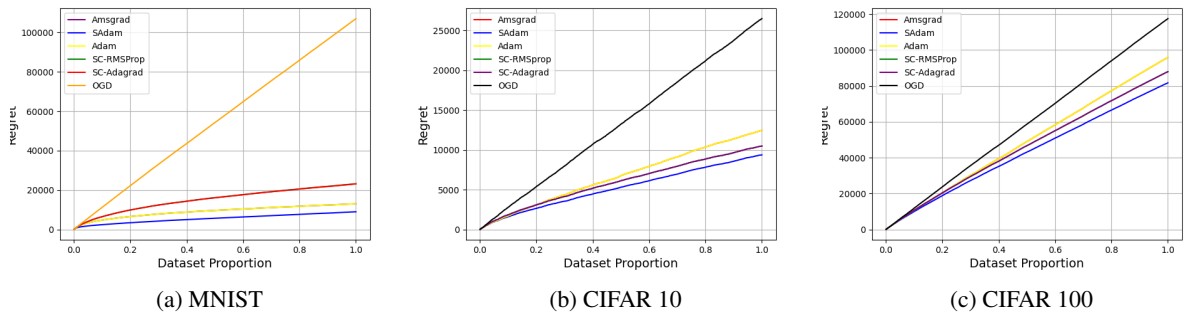

(a) MNIST      (b) CIFAR 10      (c) CIFAR 100

Figure 4: Regret vs Data Proportion for $l2$-Regularized Softmax Regression

## 4.2 Performance on 4 Layer CNN

The 4 Layer CNN is trained using different optimizers to substantiate the second claim according to which, the performance of the algorithms is sustained even in case of non convex problems. The test accuracy and the training losses are reported against all of the datasets. The performance of SAdam is again overall the highest if only test accuracy is taken into account. The performance on CIFAR 10 and MNIST is only marginally better than the next best option which is SC-RMSProp. However the test accuracy achieved after training for a 100 epochs on the CIFAR 100 dataset exceeds the values reported in the actual paper by 2% and is significantly higher than all of the other alternatives. In case only training losses are compared, SAdam achieves the lowest values in case of MNIST and CIFAR 100, but for CIFAR 10, OGD has the lowest loss value after 100 epochs. Amongst the other optimizers, SC-RMSProp consistently scores better metrics, with both Adam and AMSGrad again having comparable values following closely behind.

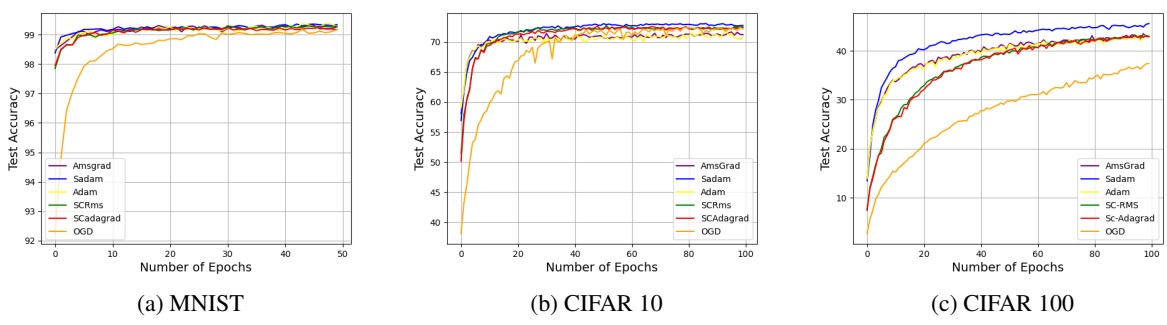

(a) MNIST      (b) CIFAR 10      (c) CIFAR 100

Figure 5: Test Accuracy vs Number of Epochs for 4 layer CNN

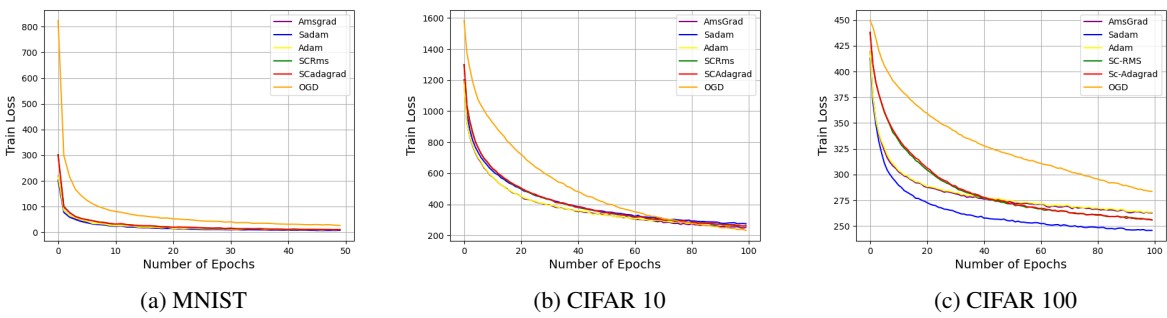

(a) MNIST       (b) CIFAR 10       (c) CIFAR 100

Figure 6: Training Loss vs Number of Epochs for 4 layer CNN

## 4.3 Performance on ResNet-18

For this experiment, like the authors, we run the experiment 10 times, and report the mean aggregate values across all runs. In our results, AmsGrad gives the best results on the test set after 100 epochs, although SAdam does appear to give the best performance after around 30 epochs, before stagnating at the same accuracy levels, since it converges to a 0 at that point, and keeps taking random jumps from that point onwards. OGD suffers from this phenomenon the most, and thus has uneven jumps sporadically throughout its learning phase. However, it must be noted that all the optimizers that take advantage of the convexity, retain their faster convergence property while training on ResNet as well.

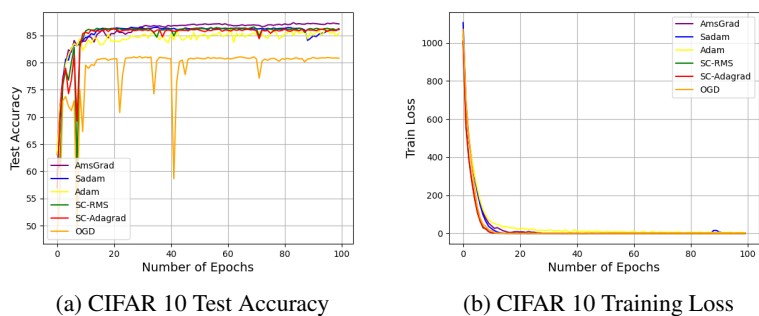

(a) CIFAR 10 Test Accuracy       (b) CIFAR 10 Training Loss

Figure 7: Peformance on Optimizers on CIFAR-10 trained on the ResNet-18 architecture

## 4.4 Multi Layer LSTM Language Model

In the language domain, Stochastic Gradient Descent(SGD) when used in conjunction with Momentum gives best results on most tasks. For both our regularised and non regularised models, the same observation ensued, with Adam performing considerably worse than the SGD Momentum combination with perplexities on the test set being higher by a factor of 5 for Adam. The performance of SAdam almost mirrored that of Adam, with marginal differences in the final perplexities as can be seen in the table below.

| Optimizer | Regularised Model | | Non-Regularised Model | |
| --- | --- | --- | --- | --- |
| | Validation | Test | Validation | Test |
| SGD+Momentum | 100.7 | 103.1 | 122.4 | 125.1 |
| Adam | 340.2 | 350.9 | 330.8 | 332.2 |
| SAdam | 343.2 | 350.7 | 331.7 | 331.9 |

Table 4: Perplexity Results on the PennTree Bank Dataset

# 5 Discussion

SAdam as shown in all of the experiments above, has the property of converging faster specially so in convex functions, but also to a certain extent in non-convex environments. Even though, the power of the algorithm diminishes with an increase in the complexity of the non-convex functions, SAdam still performs better that Adam and AmsGrad only falling short behind SC RMSProp in those cases. Certainly its performance in language tasks show that as an optimizer its utility is not domain agnostic and might suffer heavily in some other non-vision tasks. However since the performance of SAdam mirrored that of Adam in the same, any domain in which Adam has been shown to work well, working with SAdam should be taken into consideration as well since the faster convergence rates would at the very least enable quicker baseline evaluation checks of deeper networks, leading to reduced usage of hardware acceleration, while choosing amongst many model architectures.

## 5.1 Strength

Whilst writing our code, we chose to build everything from scratch in a different framework, taking the algorithms mentioned in the base papers without letting any bias creep in from the source code. We also in addition, conducted the experiments with various seeds, and the overall results remained the same. We would like to point out that, our code also serves as an open source contribution for all of the optimizers against which comparisons of SAdam were drawn, none of which except for Adam and AMSGrad had a publicly available PyTorch implementation, at the time of writing this report. We do not provide the link here, to respect the integrity of the double blind review process, however our code is available for public access on GitHub.

## 5.2 Weakness

It must be noted that although SAdam performs the best in deeper networks in most of the aforementioned experiments, there is no theoretical foundation for this empirical finding. Any attempt to do so warrants a sophisticated level of understanding of the calculus involved in the optimization mathematics and consequently our ability to address this concern was limited.

## 5.3 What was easy

The source code provided by the authors worked in one go, without modifications of any kind required. Even, when the base framework was changed to PyTorch, all the empirical experiments made in the original paper, were relatively easy to setup, right from procuring all the 3 datasets to the selection of the hyperparameters, owing to clear mention of all such variables in the paper. The algorithms itself, for each of the optimizers were clearly explained, and the translation of the algorithm to code was seamless. In addition, the low runtime for each individual run meant that, no additional hardware was required, and training on Google Colab was sufficient for each experiment given in the paper.

## 5.4 What was difficult

As mentioned before, implementing the additional optimizers in PyTorch compunded the work at hand, since to prove the efficacy of SAdam itself, a comparison with the state of the art optimizers with the same environmental variables at play was necessary. Besides this, the regret calculation experiment for L2-regularized Logistic Regression took much more time than expected owing to the contradictions in the proposed values of mini-batch size for each of the datasets, by the authors of SC-RMSProp and SAdam. Although SAdam works better for a far wider range of possible mini-batch sizes, the performance of SC-RMSProp increases, and eventually even surpasses that of SAdam if the mini-batch size is decreased substantially. The authors of SC-RMSProp themselves suggest a mini-batch size of 1, which therefore delivers the best of results for their alogrithms itself. For most real world applications, a per example based approach to learning is more likely to cause unstable optimization, and a sufficiently large mini-batch size leads to a smoother convergence, therefore holistically SAdam should be a preferred choice in a convex setting.

## 5.5 Communication with original authors

At the time of the launch of the Reproducibility challenge source code for the paper had not been shared on the open review portal. However with our first email correspondence, they shared their codebase promptly. The discrepancies in the recommended value of the mini-batch sizes proposed for the different algorithms were discussed with the authors, and they clarified their reasoning behind choosing of a different batch size. Besides this, the authors answered our minor queries with regards to the source code as well, and were supportive throughout the entire reproducibility process.

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
