# OpenReview forum: "[Reproducibility Report] SAdam: A Variant of Adam for Strongly Convex Functions"
_ML_Reproducibility_Challenge/2020 — Reject_

### Official Review · AnonReviewer3 · 2021-03-02
**nice report**

**Rating:** 7
**Confidence:** 3

**Review:**

Reproducibility Summary : The report has this summary that contains major findings.
Scope of reproducibility: The report states the scope clearly, and follows it.
Code: The authors reproduced the code in PyTorch.
Communication with original authors: Fair communications are mentioned, but I am not sure whether the original authors have evaluated the results in this report.
Hyperparameter Search: It is mentioned in the report that hyperparameters were varied, but not all results are included in the report.
Ablation Study: The ablation study is not comprehensive.
Discussion on results: The report discusses the state of reproducibility of the original paper, and mentions the easy parts and difficult parts. The numerical results of this report are consistent with those in the original paper. Only the results on ResNet-18 deviate.
Recommendations for reproducibility: It seems the report does not discuss a lot on how the original authors can improve reproducibility.
Results beyond the paper: The experiments in the report are almost the same as those in the original paper.
Overall organization and clarity: The overall quality is good.


**Familiar With The Original Paper:**

I have not read the original paper

**Reproducibility Summary:**

Report has summary

---

> ### Author Response · Authors · 2021-03-22
> **Hyperparameter Grid Search and LSTM training on PennTreeBank Dataset added**
>
> Greetings, We would like to thank the reviewer for taking the time to review our submission and for their constructive feedback. We are glad that you found the report good enough to be accepted.
>
> > Communication with original authors: Fair communications are mentioned, but I am not sure whether the original authors have evaluated the results in this report.
>
> Yes, they have reviewed the results of the earlier experiments and they had a major hand in the suggestion of the new experiments that have been added.
>
> > Hyperparameter Search: It is mentioned in the report that hyperparameters were varied, but not all results are included in the report.
>
> We have added the Hyperparameter Grid Search results for the L2-Regularized Softmax Regression experiment. The variance of results with the optimizer hyperparameters was fairly low and the most important hyperparameter seems to be the learning rate. $\beta_1$ is chosen from {$0.0, 0.9$}, $\beta_2$ is chosen from {$0.9, 0.99, 0.999, 0.9999$} and the learning rate $\alpha$ is chosen from {$0.1, 0.01, 0.001, 0.0001$}.
>
> > Ablation Study: The ablation study is not comprehensive.
>
> >  The experiments in the report are almost the same as those in the original paper.
>
> We hope we have improved our submission in this aspect by the addition of the experiment where both 2-layer Regularized and Non-Regularized LSTMs are trained on the PennTreeBank dataset and the perplexity is reported. We also compare the model trained with SAdam to a model trained with Adam and one trained using SGD with Momentum. SGD with Momentum is the best as expected and SAdam is very close in performance to Adam but falls just short. LSTMs are relevant in this case as they contain many matrix multiplication the optimization surface becomes flat and strays away from being convex[1]. Hence SAdam which is strictly meant for convex surfaces does not perform best here.
>
> > It seems the report does not discuss a lot on how the original authors can improve reproducibility.
>
> The reproducibility of the original code by the authors was already very good. The code is available for everyone to use, and each of the experiments can be run, after going through the code once. A possible addition of a good Readme file, would however have been a massive help, as some of the time to go through the code would have been reduced, an issue that we have taken into consideration, and rectified in our Pytorch implementation of the codebase, providing exact steps to reproduce all of the results in each of the experiments.
>
> [1] Sequence Modelling: Recurrent and Recursive Nets (Chapter 10) from Deep Learning by Ian Goodfellow, Yoshua Bengio, Aaron Courville

---

### Official Review · AnonReviewer1 · 2021-03-02
**A decent reimplementation and confirmation of SAdam on simple frameworks**

**Rating:** 5
**Confidence:** 4

**Review:**

The report clearly presents its scope and reimplements SAdam on pytorch where they compare their implementation with the version obtained from the authors which is in tensorflow. The report has multiple positive aspects:
- communication with the authors
- reimplementation of the code
- clear reproduction of the original paper
- clear description of the problem and the presentation of the report
- additional experiments on the resnet model

Major shortcomings of the paper include:
- lack of a range of hyperparameter search for the robustness of results
- testing the algorithm in contexts other than simple vision problems

The area of algorithmic improvements is a tricky subject where it is hard to confirm with strong confidence that the results are robust across wide variety of hyperparameters, and confirming the validity of the results in a way that translates to different structures of problems. I think one of the strongest points a reproducibility can make is to show that this is the case. That said, the present report does a great job at reproducing the core components of the investigated paper; I encourage the authors to extend their hyperparameter search for robustness and comparisons as well as applying the algorithm to wider (different) sets of optimization problems.

**Familiar With The Original Paper:**

I have read the original paper

**Reproducibility Summary:**

Report has summary

---

> ### Author Response · Authors · 2021-03-22
> **Hyperparameter Grid Search and LSTM training on PennTreeBank Dataset added**
>
> Greetings, We would like to thank the reviewer for taking the time to review our submission and for their constructive feedback. We hope the additions listed below will change your mind regarding the rating of the report. We have tried to address both shortcomings as best as we could in the span of this one week.
>
> > lack of a range of hyperparameter search for the robustness of results
>
> We have a added a section with Hyperparameter Grid Search results for L2-Regularized Softmax Regression experiment. The variance of results with the optimizer hyperparameters was fairly low and the most important hyperparameter seems to be the learning rate. $\beta_1$ is chosen from {$0.0, 0.9$}, $\beta_2$ is chosen from {$0.9, 0.99, 0.999, 0.9999$} and the learning rate $\alpha$ is chosen from {$0.1, 0.01, 0.001, 0.0001$}.
>
> > testing the algorithm in contexts other than simple vision problems
>
> To address this we have added the experiment where both 2-layer Regularized and Non-Regularized LSTMs are trained on the PennTreeBank dataset and the perplexity is reported. We also compare the model trained with SAdam to a model trained with Adam and one trained using SGD with Momentum. SGD with Momentum is the best as expected and SAdam is very close in performance to Adam but falls just short. LSTMs are relevant in this case as they contain many matrix multiplication the optimization surface becomes flat and strays away from being convex[1]. Hence SAdam which is strictly meant for convex surfaces does not perform best here.
>
> We would have loved to train more complex image models such as GANs and report results on other complicated tasks such as Neural Machine Translation and Reinforcement Learning but we were not able to due to compute restrictions.
>
> [1] Sequence Modelling: Recurrent and Recursive Nets (Chapter 10) from Deep Learning by Ian Goodfellow, Yoshua Bengio, Aaron Courville

---

### Official Review · AnonReviewer2 · 2021-03-08
**Good reproducibility report**

**Rating:** 7
**Confidence:** 4

**Review:**

This manuscript provides an pytorch implementation and reproducibility report of "SAdam: A Variant of Adam for Strongly Convex Functions". Most of the claims are consistent with the original paper. Experimental results include L2 regularized softmax regression, 4 layer CNN, ResNet 18 on MNIST, CIFAR10 and CIFAR100 datasets with exhaustive hyperparameter tuning. It would be interesting to see the results on ImageNet.

**Familiar With The Original Paper:**

I have read the original paper

**Reproducibility Summary:**

Report has summary

---

> ### Author Response · Authors · 2021-03-22
> **ImageNet Training not possible due to compute restrictions**
>
> Greetings,
> We would like to thank the reviewer for taking the time to review our submission and for their constructive feedback. We are glad that you found the report good enough to be accepted.
>
> > It would be interesting to see the results on ImageNet.
>
> This was not possible for us in this one week span due to compute restrictions and so we regret to inform that we have not been able to add that experiment.
>
> We however have added Hyperparameter Grid Search results for the L2-Regularized Softmax Regression and results for LSTMs trained on the PennTreeBank dataset with SAdam and compared that with LSTMs trained with Adam and SGD with Momentum.

---

### Decision · Program_Chairs · 2021-03-31

**Decision:**

Reject

**Comment:**

There is a lack of a range of hyperparameter search for the robustness of results and in testing the algorithm in contexts other than simple vision problems.

---

> ### Author Response · Authors · 2021-04-03
> **Reasons cited for rejection were addressed in the revised submission**
>
> There seems to be some discrepancy in the reasons cited for rejecting the paper, since all the points mentioned for rejection had all been individually addressed in the revised edition of the report submitted on the 21st of March, 2021.
> > There is a lack of a range of hyperparameter search for the robustness of results
>
> In the revised edition we added a section with Hyperparameter Grid Search results for L2-Regularized Softmax Regression experiment, and reported the results over a combination of 20 hyperparameter configurations, with the parameter  α  (learning rate) chosen from amongst 4 potential values of {0.1, 0.01, 0.001, 0.0001}, β1  from amongst  the values of {0.0, 0.99}   and β2  from {0.9, 0.99, 0.999, 0.9999}.
>
> > testing the algorithm in contexts other than simple vision problems.
>
> Additionally we added an experiment where both 2-layer Regularized and Non-Regularized LSTMs were trained on the PennTreeBank dataset and the perplexity for each was reported. We compared the model trained with SAdam to a model trained with Adam and SGD with Momentum respectively. A detailed analysis of both of these changes were also mentioned in the report.
>
> It would help greatly, if you could look into the matter, and clear out the confusion.